# Effects of Sports, Exercise Training, and Physical Activity in Children with Congenital Heart Disease—A Review of the Published Evidence

**DOI:** 10.3390/children10020296

**Published:** 2023-02-02

**Authors:** Simone K. Dold, Nikolaus A. Haas, Christian Apitz

**Affiliations:** 1Department of Pediatric Cardiology and Pediatric Intensive Care Medicine, Ludwig-Maximilians University Hospital Munich-Großhadern, 81377 Munich, Germany; 2Division of Pediatric Cardiology, Children’s Hospital, University of Ulm, 89075 Ulm, Germany

**Keywords:** congenital heart disease, fitness, sports, physical activity, Fontan, training, rehabilitation, musculoskeletal functioning, motoric skills, cardiorespiratory fitness, quality of life, child, children, youth, adolescents

## Abstract

Children and adolescents with congenital heart disease (CHD) should be encouraged to adopt a physically active lifestyle, ideally by participating in sports activities at school and sports clubs. Children with complex CHD or other risk factors (for example, pacemakers, cardioverter-defibrillators, channelopathies) may, however, need specific individualized training programs. This review article summarizes the current knowledge regarding the clinical effects of sports and exercise training on CHD and its pathophysiologic mechanisms. An evidence-based approach based on a literature search, using PubMed, Medline, CINHAL, Embase, and the Cochrane Library was conducted, last completed on 30 December 2021. In studies with 3256 CHD patients in total, including 10 randomized controlled trials, 14 prospective interventional trials, 9 observational trials, and 2 surveys, exercise training has been shown to improve exercise capacity and physical activity, motoric skills, muscular function, and quality of life. Sports and exercise training appears to be effective and safe in CHD patients. Despite being cost-efficient, training programs are currently scarcely reimbursed; therefore, support from healthcare institutions, commissioners of healthcare, and research-funding institutions is desirable. There is a strong need to establish specialized rehabilitation programs for complex CHD patients to enhance these patients’ access to this treatment intervention. Further studies may be desirable to confirm these data to investigate the impact on risk profiles and to identify the most advantageous training methodology and underlying pathophysiological mechanisms.

## 1. Introduction

### 1.1. Background

During recent decades, there has been a huge change in medicine and health prevention in general to a more early and active approach towards health prevention.

It is well known that sedentary behaviors and lifestyles are associated with an increase in morbidities such as obesity, metabolic disease, and cardiorespiratory risk factors during adulthood. A more active lifestyle and health education programs could reduce some of these health problems. Nevertheless, the basis for a long and healthy life is laid in early childhood [1]. Especially in children with congenital heart defects, any acquired cardiac disease will complicate the outcome.

Current international guidelines for healthy children recommend participating in a daily minimum of 60 min of moderate-to-vigorous-intensity physical activity [2,3,4,5,6]. Physical activity includes every active movement which increases basal metabolism. Nevertheless, only a tiny number of 12% of healthy children reach this goal [7]. Recommendations for children in Germany even suggest 90 min or at least 12,000 steps per day [8]. These recommendations were not targeted at children with chronic illnesses. However, especially chronically ill children tend to be “overprotected” by parents, and even their treating doctors are often not sure what to recommend regarding physical activity [7,9,10]. This insecurity and fear lead to an even more sedentary lifestyle in children with chronic diseases, including children with congenital heart defects, and results in a growing number of cardiorespiratory risk factors, obesity, metabolic disease, and even developmental delay in early childhood. 

This topic is of special interest as most children with CHD survive until adulthood. The more sedentary behavior with the increasing age of this patient group is of special importance as these grown-up CHD patients often may drop off the radar of structured follow-ups [11,12,13,14,15,16,17,18,19,20]. Currently, there are just a few studies on this specific topic, mostly, however, with a low evidence level, a small number of participants, and a close focus on only one aspect.

Pediatric cardiologists more and more agree that there is no strict prohibition for those with CHDs to live an active life and even participate in certain sports [21,22,23,24,25]. Individual approaches and risk analysis seem to be appropriate, with health benefits clearly outweighing the doubts [26,27,28]. 

In 2013, the American Heart Association recommended physical activity for children and adults with congenital heart defects [3]. Their recommendations were based on those for healthy children and adults as a specific recommendation for this patient group is still lacking. It was even concluded that there is no evidence suggesting any restrictions apart from patients with severe rhythm disorders. They did not give any advice regarding exercise types or kinds of sports but they recommend generally less sedative behavior. 

The recently published guideline of the German Association of Pediatric Cardiology (DGPK) suggests a standardized examination protocol to ensure safety and reduce adverse events using an individualized approach dependent on the underlying condition to encourage a higher level of physical activity and participation in sports in children with congenital heart disease and reduce insecurity. This guideline underlines the current evidence that there are very few conditions that prohibit participation in sports in general [29]. 

In this guideline, sports and training are categorized according to the different approaches with either a more dynamic or a more static approach. Aerobic training approaches have a high dynamic component and result in a reduced afterload and therefore a higher volume load. In comparison, training with a static component such as weight training results in a rise in pressure. 

Depending on the underlying condition and the here from resulting problems, the best-fitting sports can be identified. As an example, patients with Fontan palliation should avoid static training because of the increase in pressure, which can lead to a reduction in cardiac output in these patients, depending on passive lung perfusion. 

Another important factor the guideline is focusing on is anticoagulation. Depending on the risk of injuries, especially head injuries, some contact sports should be avoided. 

A special position is held by children with CHD participating in competitive sports. If they decide to continue, the DGPK recommends periodic reevaluations with a special focus on ventricular function, pulmonary artery pressure, the dimensions of the aorta, arrhythmias, and oxygen saturation to detect early changes in risk factors.

Exercise intolerance or sedative behavior may put children with congenital heart disease at an additional high risk to develop comorbidities such as diabetes, obesity, anxiety, and depression [12]. Additionally, there are relevant changes during childhood and adolescence which also have an impact on the cardiopulmonary and musculoskeletal systems. Cardiac rehabilitation can improve the exercise performance of children with CHD. This improvement is mediated by an increase in stroke volume and/or oxygen extraction during exercise performance. The routine use of formal cardiac rehabilitation programs may greatly reduce the morbidity of complex CHD and improve long-term outcomes [30].

Regarding the large number of positive aspects related to an active lifestyle with almost no adverse effects, parents and children with CHD should be even more motivated to participate in regular sports and exercise. This will help them to become a bit more integrated into their peer groups, confident, and independent with a rising quality of life. Physical activity starts with the way to school. Children who walked or used their bicycles to go to school tended to have a higher level of physical activity in general [31]. There are also several trials focusing on ways in which healthy children could be motivated to be more active [32,33,34,35,36,37,38,39,40,41,42,43,44,45,46,47,48]. A positive effect had free play times during school, colorful playgrounds, and a playful approach in general [32,33,34,35,36,37,49,50]. In Germany, there was a trial evaluating the effect of a trampoline on physical activity (PA) levels [51]. Trials focusing on the effect of church leisure events or summer camps did not show a lasting effect regarding PA levels [52]. 

The different activity levels are defined by Takken et al. They also defined the static and dynamic impact of sports and activity categories, giving a hint of which category is more useful for CHD patients, according to the type of heart defect and the severity of their illness [27].

### 1.2. Pathophysiologic Mechanisms of Sports and Training in CHD

The exercise limitation in CHD patients might be multifactorial. It can be caused by ventricular dysfunction, chronotropic incompetence, ventilatory abnormalities, and skeletal muscle dysfunction. The mechanisms of exercise intolerance are complex, likely including respiratory muscle weakness, dynamic hyperinflation and mechanical constraints, poor skeletal muscle and cerebral oxygenation, hyperventilation, and enhanced sympathetic drive. Likewise, exercise training improves the function of different body organs such as the heart, lungs, and skeletal muscles. 

The mechanisms of improved hemodynamics and exercise capacity by exercise training in individuals with CHD remain incompletely understood [Figure 1]. Increased cardiac output at rest and at maximum exercise may be explained either by a decreased afterload of the ventricles, either a muscular or molecular effect on the blood vessels with a reduction in afterload and/or pulse wave velocity, or a direct myocardial training effect. In addition, improved exercise capacity is at least in part to be explained by improved skeletal muscle function, but there are, however, no reported direct measurements of improved diffusional muscle oxygen uptake by exercise training.

Furthermore, a combined effect on different molecular pathways and organs is likely to be the pathophysiological underpinning of the improvement associated with exercise training in individuals with CHD. Further research is needed to elucidate the relevance of each of these mechanisms. It is also of great interest whether exercise training leads to epigenetic changes.

### 1.3. Objectives

The aim of this review is to give a broad overview of the effects of higher PA levels, participation in sports, and training/rehabilitation programs with a focus on children and adolescents with CHD. It also evaluates if there is a trend in which training programs or approaches work better. 

## 2. Methods

### 2.1. Search Strategy

An internet-based literature search for relevant, original research articles on the effects of sports, fitness, and physical activity on the effect of cardiorespiratory fitness, reductions in morbidities, muscle function, and quality of life in children with CHD was conducted (last completed 30 December 2021). We searched PubMed, Medline, CINHAL, Embase, and the Cochrane Library using the search terms:

((((((((((((congenital heart defect AND sport AND children)) AND (Congenital heart disease AND sport AND children)) AND (congenital heart defect AND fitness AND children)) AND (Congenital heart disease AND fitness AND children)))) AND (congenital heart defect AND physical activity AND child)) AND (congenital heart disease AND physical activity AND child)) AND (congenital heart defect AND physical activity AND cardiorespiratory fitness AND children)) AND (congenital heart disease AND physical activity AND cardiorespiratory fitness AND children)))

→ 134 articles

((((((((congenital heart defect AND exercise training AND child)) AND (congenital heart defect AND exercise training AND children)) AND (congenital heart disease AND exercise training AND child)) AND (congenital heart disease AND exercise training AND children)) AND (congenital heart disease AND exercise training AND child AND cardiorespiratory fitness)) AND (congenital heart disease AND exercise training AND children AND cardiorespiratory fitness)) AND (congenital heart defect AND exercise training AND child AND cardiorespiratory fitness)) AND (congenital heart defect AND exercise training AND children AND cardiorespiratory fitness)

→177 articles

Physical activity interventions for children with congenital heart disease

→ 1353 articles

((((((((exercise training AND congenital heart disease AND child)) AND (exercise training AND congenital heart disease AND children)) AND (physical activity AND congenital heart disease AND child)) AND (physical activity AND congenital heart disease AND children)) AND (physical activity AND congenital heart disease AND youth)) AND (physical activity AND congenital heart disease AND adolescent)) AND (exercise training AND congenital heart disease AND youth)) AND (exercise training AND congenital heart disease AND adolescent)

→ 4629 articles

This search produced a total of 6293 articles. The articles were thoughtfully sorted, and duplicates were removed. Afterwards, the references were searched for additional literature.

### 2.2. Inclusion Criteria

We included all kinds of studies evaluating the effect of either PA, sports, exercise training, or rehabilitation programs with a focus on PA levels on children and adolescents with CHD. The articles must be available as full texts. 

### 2.3. Exclusion Criteria

We excluded studies not relevant to the pediatric population, letters to the editor, review articles or commentaries, and articles only available as abstracts. Studies that were not yet completed were also excluded, as well as reviews and meta-analyses.

### 2.4. Data Extraction

A total of 40 studies matched our inclusion criteria [Figure 2]. Those were further grouped into 29 studies examining any kind of intervention with its effects, and 11 studies describing the effects according to questionnaires.

### 2.5. Level of Evidence 

Clinical trials were graded according to the standardized Level of Evidence (LOE) I-IV [36,37].

## 3. Results

The studies were published between 1990 and 2021 [Table 1 and Table 2]. The number of participants varied from 7 to 477. The follow-up time in the intervention group also differed, ranging from 8 weeks to 24 months. Longmuir et al. used earlier gathered data and performed a 5-year follow-up with them [53]. Additionally, the effects underwent different examinations. The age group varied between toddlers and young adults. Therefore, the transmission of the results to patients with CHD, in general, is not rational and should be considered carefully. 

### 3.1. Motoric Skills

It is well known in healthy children that regular exercise improves motoric skills such as climbing, balance, and coordination.

Stieber et al. had the youngest examined age group of children with CHD with a group of toddlers, the youngest aged 12 months, after surgical correction or palliation. This study solely focused on the development of motoric skills, as these children, especially those after Fontan palliation, are significantly delayed in their motoric skills in comparison to their healthy peers. Those toddlers underwent a 10-week play-based, parent-delivered rehabilitation program with a follow-up examination after completing the 10-week program with daily training of 20 mins. Patients after Fontan palliation reached significant improvements in their motoric skills and reached values comparable to their age group [74]. 

Müller et al. showed similar results in their pilot study group of 14 CHD patients aged 4–6 years. In this trial, the impaired children significantly improved their motoric skills after 3 months of training with a 60 min session per week [73]. 

Ferrer-Sargues et al. conducted a study on 15 adolescents (12–16 years old), with two separate focuses. In a first approach, they examined the effect of their IMPROVE project (Initiative for Monitored Pediatric cardiac Rehabilitation Oriented by cardiopulmonary Exercise testing), designed following the American College of Sports Medicine (ACSM) on peripheral muscle function. This included 2x/week sessions of 70 min for a total of 24 sessions. The sessions included endurance and strength-resistance training. The intensity was defined by the subject’s CPET results, initially aiming for a heart rate (HR) near the first ventilatory threshold (VT1) HR and displacing this target frequency progressively towards the secondary ventilatory threshold (VT2) HR or a maximal HR of 75% of their peak HR in cases where the VT2 was not available. The muscle function was tested at baseline, after completion, and 6 months after. In a second approach, they focused on respiratory muscle function. This study did not evaluate the impact on the CRF nor QoL. However, they could show an improvement regarding muscle function [55,56].

Longmuir et al. also evaluated the effect of either a parent-delivered home exercise or education program, both aiming to increase PA levels. Patients were randomized to either of these programs. Each program lasted for 12 months with a 1.5–2 h session per week. At the end of the trial, the patients showed significantly improved gross motor function and significantly increased moderate to vigorous physical activity (MVPA). These results lasted until the 2 years follow-up. This study had a relative representing a number of 61 Fontan patients, aged 6–12 years [69].

Brassard et al. studied seven Fontan patients with an 8-week training program, including an aerobic and a resistance part. They could show an effect on neuromuscular function and a strong but not significant reduction regarding the pulse wave velocity with a consecutive reduction in blood pressure [76].

Moalla et al. showed reduced muscle function in patients with CHD when performing resistance training and slower reperfusion. In another study, they randomized patients in aerobic cycling training and could afterwards show an improvement in strength and endurance. Moreover, there also was improved oxygenation and faster recovery [71,72].

In general, the included studies could show a positive effect on muscle development and motoric skills in children and adolescents with CHD, resembling the effect physical activity has on healthy individuals.

### 3.2. BMI

BMI and weight often correlate to physical activity levels in healthy children. Exercise participation leads to weight loss and a reduction in waist circumference.

Altamirano-Diaz et al. studied the effect of biweekly fitness and nutrition counseling delivered via smartphone over a period of 12 months in 34 obese CHD patients aged 7–17 years. The sessions lasted 30 min each. The groups were divided into CHD-operated and not-operated groups. Waist circumference significantly decreased by 2.61 cm at 6 months and by 2.25 cm at 12 months in the operated group. A significant increase in lean body mass was observed in both groups. No differences were seen regarding blood tests and CPET [62]. 

In a study conducted by Fredriksen et al. which included a 5-month training program in 10–16-year-old patients with CHD, compared to a control group without training. They observed weight gain in the control group without training [79].

There are also other studies that assessed BMI and PA levels with questionnaires. They could show a significant correlation between PA levels and BMI in patients with CHD. With a higher level of activity, the BMI decreases [80,84]. 

Other studies showed a strong correlation between reduced PA levels in CHD patients and a higher percentage of obese patients [91,92,93]. 

Regarding the effects on BMI and body weight, physical activity has also a similar effect on patients with CHD as it has in healthy individuals. A higher level of physical activity is associated with a lower body weight and therefore a lower BMI, as well as a reduced waist circumference.

### 3.3. Cardiorespiratory Fitness

Cardiorespiratory fitness is known to be associated with morbidity and mortality; therefore, this part is of special interest regarding the effects of interventions. 

Sutherland et al. assessed the effect of an 8-week home vs. hospital-based training program in 17 Fontan patients aged 12–19 years of age. The program was performed according to a standardized program with 2x1 h training sessions per week. The training followed a structure of a 5–10 min warm-up, followed by an equally divided 20–30 min aerobic and resistance part, following a 5–10 min cooldown. The effect on the cardiorespiratory fitness was assessed after completing this 8-week training program with a CPET examination and a 6MWT. Both training programs showed a significant improvement regarding oxygen consumption, peak pulse as well as the distance committed during 6MWT. Additionally, the patients had to fill in a questionnaire evaluating their quality of life. This also showed a significant improvement after completing the training programs [61].

Kroll et al. recently published their results in 25 Fontan patients completing a home-based year-long cardiac rehabilitation program. This program included four in-person visits of an interdisciplinary team, including a cardiologist, a physical therapist, an occupational therapist, a psychologist, and an exercise physiologist every 3–6 months as well as wearing an activity monitor. The effects of the program on the exercise capacity were assessed using the Progressive Aerobic Cardiovascular Endurance Run (20-meter shuttle test run). Additionally, the patients had to fill in several questionnaires. The results showed a significant increase in the median completed shuttles from 5 to 10. The quality of life was not significantly improved regarding the patients’ forms. Conversely, the parents stated that there was a significant improvement. Eleven of the twenty-five included patients had an additional diagnosis, for instance, attention deficit hyperactivity disorder (ADHD) [57].

Jacobsen et al. conducted a pilot study in 14 Fontan patients using a 12-week moderate/high-intensity home-based cardiac physical activity program, including three formalized in-person exercise sessions at weeks 0, 6, and 12. The exercise sessions were 45 minutes with a mixture of dynamic and static exercises. Additional data were collected with an activity monitor the children had to wear during the study time. For the evaluation, there were also CPET and Shuttle Test runs used. The objective measurements regarding endurance and cardiorespiratory fitness improved significantly. They also used questionnaires to evaluate the effects on quality of life. In this trial, the statements of the parents and children brought controversial results [64].

Moalla et al. studied 17 patients with CHD aged 12–16. Nine of them were randomized to a 12-week home-based training program with three 1 h sessions per week. The adolescents used a cycle ergometer and a pulse monitor for the duration of the study. The training program included a 10 min warm-up, followed by a 45-minute interval training session alternating with 10 min active and 5 min passive periods on the cycle ergometer. The active periods aimed to achieve an individualized target heart rate according to the HR achieved at VT in a previously conducted CPET. The other eight patients were randomized to a passive control group as well as fourteen healthy peers. The results were measured using CPET and 6MWT. After the 12 weeks of training, there could be seen a significant improvement regarding cardiorespiratory fitness in the training group [71,72].

Callaghan et al. studied 163 patients with CHD. They were randomized in an intervention group, starting with a 1-day education session followed by a 4-month training program with an individualized written exercise plan. An evaluation regarding cardiorespiratory fitness was conducted using CPET, and activity was monitored using an accelerometer. They could show a significant improvement in peak exercise capacity in the intervention group as well as a trend towards increased daily activity levels [54].

Brassard et al. studied seven Fontan patients with an 8-week exercise program either at home or in the hospital. The training program was split into an aerobic and a resistance part. The conducted CPET before and after the exercise training showed no significant benefit. Nevertheless, they showed some changes in muscle function and blood pressure reduction [76].

In the multicenter trial of Dulfer et al. and Duppen et al, there could be seen a significant improvement in cardiorespiratory fitness (CRF) with an improvement in VO_2_max and O_2_-pulse in patients with the tetralogy of Fallot, whereas this effect could not be verified in Fontan patients [67,68].

Hedlund et al. showed similar results in 30 Fontan patients after a 12-week endurance training at a submaximal level. Healthy controls increased their CRF. Fontan patients did not increase their CRF, but they increased the distance in the 6MWT and their quality of life [63].

Fredriksen et al. achieved a significant increase in VO_2_max in 129 patients who underwent a 5-month exercise training program at home or in a center [79].

Meyer et al. couldn´t see any improvement after a 24-week web-based exercise training program. The program was divided into 3 × 20 mins per week. The exercise sessions included child-friendly video instructions and demonstrations of the different exercises. It also included a virtual training partner, and the exercise was performed simultaneously with the video. Participation was assessed with tracking, and regular reminders were sent. Afterwards, the HRPF was assessed in accordance with the FITNESSGRAM, which is a standardized test to assess HRPF with exercises such as curl-ups, trunk lifts, and push-ups. There was no improvement found after the completion of the program [58].

Opocher et al. studied 10 Fontan patients and could measure under submaximal endurance an increased O_2_ pulse and an increase in VO_2_ [78].

In summary, different interventional programs seem to have a positive effect with an improvement in VO_2_max and therefore a measurable increase in cardiorespiratory fitness in almost all patients with CHD. Nevertheless, this seems not to be achievable in Fontan patients. In these patients, there seems to be no increase at the maximum exercise test, but Opocher et al. could show an improvement at a submaximal level [78], as well as Brassard et al., which could show a reduction in pulse wave velocity and therefore blood pressure [76]. Therefore, in Fontan patients, the benefits seem to be found on another level.

### 3.4. Physical Activity 

Physical activity is the crucial point in everyday activity. It decreases with age in healthy individuals.

Longmuir et al. assessed the effect of a 3-month home exercise program on the level of physical activity using a questionnaire in patients with CHD. The program was not described in detail but included endurance activities such as jogging. A control group of healthy peers was questioned. They showed a significant improvement in PA levels, even after a 5-year follow-up [53]. 

Dulfer et al. and Duppen et al. examined 93 TOF/Fallot patients aged 10–25. The intervention group was randomized to a 12-week standardized exercise training program with 1 h sessions three times a week. The training started with a 10 min warm-up, then 40 min aerobic dynamic cardiovascular training, and ended with a 10 min cooldown. The results were assessed via questionnaires and accelerator data. Exercise training decreased passive, sedentary leisure-time spending; active leisure time-spending was not reduced. The patients seemed to take part in their regular active activities. Nevertheless, there could not be seen an increase in PA in general [66,67,68].

Fredrikson et al. measured PA in 129 CHD patients and a control group before and after either training or no intervention. They showed a significant increase in PA 1–2 weeks after the completion of the training [79].

Klausen et al. used an eHealth application for a period of 1 year to assess the effects of PA and CRF. The app reminded the participants regularly to move and was set up in a playful way with achievements, but nevertheless, the app was hardly used, and they could not achieve any significant changes. They concluded that this approach was not useful [65].

Lopez et al. randomly measured physical activity in children with CHD without any intervention program. They showed a correlation between PA and lower pulse wave velocity. They concluded that physical activity seems to have an impact on vascular function and stated that a higher level of physical activity should be engaged in these children [60].

Morrison et al. included a sports psychological part to improve the long-lasting benefit of the training program and succeeded with an improvement in physical activity [70]. 

In summary, most of the conducted training programs and interventions had a positive effect on overall physical activity, even beyond the study period. 

### 3.5. Quality of Life and Other Questionnaires

Another important point, which is known to be lower in patients with CHD, especially with aging, is quality of life. Therefore, it is also of importance to evaluate the effect of sports and physical activity on quality of life, as this might lead to better well-being and a reduction in psychological illnesses such as depression.

Dulfer et al. also assessed a quality-of-life questionnaire, which showed, especially for the ages of 10–15 years, a significant improvement in overall quality of life. Furthermore, they reported a significant cognitive improvement. Parents also reported an improvement in social interaction, whereas they could not find an improvement regarding emotional and behavioral problems [66,67,68]. 

Hedlund et al. showed in Fontan patients an increase in quality of life after a training program, even if they did not increase their CRF, whereas healthy individuals did increase their CRF but did not subjectively increase their quality of life [63]. 

Blais et al. studied 11 children with CHD aged 7–10 years who participated in a 10-week program, including once-weekly multi-sports programs. Each lesson focused on a different ball sport. The participating children were asked to fill in questionnaires and attend group sessions. The results showed that enjoyment of physical activity is a primary source of motivation. Therefore, intrinsic motivation seems to be crucial [59]. 

Fredriksen et al. also showed a significant reduction in withdrawal and somatic complaints in the exercise intervention group [79].

Moons et al. could show an improvement and a lot of benefits during the attendance of the camp but not afterwards [77].

Kroll et al. studied 25 patients with CHD and evaluated their quality of life with several questionnaires conducted by patients and parents. The results between parents and patients differed widely. Whereas the parents noticed an overall improvement, including in psychosocial aspects, the patients only reported cardiac-related improvements in quality of life according to the questionnaires [57].

Meyer et al. could not detect an improvement in quality of life after a 24-week web-based exercise training program [58].

Sutherland et al. conducted an 8-week home vs. hospital-based training program in 17 Fontan patients. They could also show an improvement in quality of life [61].

Jacobsen et al. studied 14 Fontan patients and could show an objective improvement in CRF after the intervention. Nevertheless, there was no improvement in quality of life according to the child’s questionnaire, whereas parents stated a significant improvement [64].

As stated above, there are different results regarding perceptions of the influence the stated exercise programs had on the patients’ quality of life. There are controversial reports in the available literature regarding the patients’ own judgement for the improvement of their quality of life as well as in comparison to their parents’ reports. In the above-mentioned studies, often, parents stated a significant improvement regarding quality of life and psychosocial and behavioral issues, which was not recognized the same way by the patients themselves.

This is discussed to be a matter of self-awareness, which seems to be a difficult matter in children and adolescents, as this self-awareness has to be learned. 

According to this subjective awareness, other studies comparing the results of questionnaires or accelerometer data of CHD patients with healthy pears also showed differing results. As reported by those studies, children and adolescents with CHD can have similar physical activity levels, but they also can vary sharply as well. Maybe these confusing results are due to the different severities of the examined CHD population as well as their environment.

In conclusion, quality of life seems to be improved, and psychosocial and behavioral issues seem to be reduced by a higher level of physical activity and participation in sports. Even if patients are not immediately aware of this overall huge positive effect, their surrounding peers and family recognize early positive effects.

## 4. Discussion

The reviewed studies suggest clearly that participating in sports and exercise/rehabilitation programs, or simply the increase in physical activity, are not only safe but also beneficial for patients with CHD.

The level of evidence nevertheless is moderate as there is only a limited number of clinical trials with a small sample size of patients, focusing on children and young adults with CHD, which leads to controversial and confusing results. Regarding the widely different conditions gathered under the term “congenital heart disease”, general transferability is almost impossible. 

In summary, it can nevertheless be stated that an increase in a physically active lifestyle results in a number of positive adaptions of the body which may lead to a reduction in morbidities and even influence the mindset of the individual positively.

Regarding the relatively small number of patients with CHD living in the same area and having a comparable range of age, a home-based training and rehabilitation program seems to be the most convenient approach to start and retain families. Those programs can use modern media such as online video conferences or app-supported training, even if the trials gathered controversial results using this approach. 

Nevertheless, safety aspects are still important issues regarding, for instance, cardiac and orthopedic conditions. Therefore, a thoughtful examination, including the measurement of blood pressure, ECG, clinical examinations, and a detailed past medical and family history should be conducted. In addition, echocardiography, CPET, and Holter-ECG are useful to ensure safety and exercise tolerance. Furthermore, the trainer should supervise the correct exercise performance.

The used exercise programs also differed widely with some using a combination of aerobic and resistance exercises and others focusing on either or.

Some trials seemed to be more training-orientated and performed a CPET at the beginning to measure thresholds and use them to control training with target heart rates in the intended range [30,55,56,61,66,76,77,78,79]. As in athletes, they trained their patients to focus on their heart rates measured by, for instance, an accelerator or a smartwatch. This seems a good approach to increase the independence of patients with congenital heart defects, which correlates with normalization and an increase in quality of life.

Maybe we should rather aim to increase this independence even more, inspired by the abovementioned approach. After supervised home-based or institutional training to assure patients and their families, healthcare providers should aim to include those children in regular sports clubs, matching their interests. This might be an approach to increase intrinsic motivation and create peer groups, which has been shown to be crucial to increase PA in general. 

Another approach that was used in some of the trials, which is also used by athletes, is the Borg Scale, which assesses subjective exhausting levels and can help to assume which training intensity is equivalent to different levels in CPET. 

Furthermore, it should be noted that questionnaire results between parents and children differed largely in some studies. This may be due to the gathered security of parents achieved by knowing their children participated in a safe program and gained a step towards normality. Insecurities in parents often result in the overprotection of these children. Healthcare professionals are also insecure due to lacking guidelines and recommendations for this patient group; they often also tend towards overprotection. Therefore, it is crucial to inform especially healthcare professionals to encourage participation in sports. Healthcare prescriptions also have shown a beneficial effect. The recent guideline of the DGPK can therefore be an additional help for healthcare professionals [29].

For children in general, it is important to be led at an early age to a healthy, physically active lifestyle, and the latest data on adults with congenital heart defects showed a huge dropout rate during transmission from pediatricians to adult cardiology [16]. Patients with CHD showed a reduced quality of life decreasing with age [92]. 

Therefore, the maintenance of the interest in gathered routines should be one of the most challenging aspects for further research. 

### Limitations

This review has limitations. There is only a limited number of trials, focusing on children with congenital heart disease. As abovementioned, beyond that, the reviewed studies have limitations, such as small sample sizes, varying levels of evidence, and different approaches regarding interventions and measuring results. 

Moreover, the search was conducted by only one researcher; therefore, the false exclusion of articles cannot be fully dismissed. Nevertheless, the methods and results were approved by the other researchers.

Furthermore, academic disciplines and medical science are rapidly developing fields. Therefore, there might be other studies completed in the meantime, as the underlying search was last completed at the end of 2021.

## 5. Conclusions

In summary, PA, participation in sports, and exercise training seem to have a wide range of positive impacts, starting with improving motoric skills and muscle function, bringing health benefits, and, finally, having a huge impact on quality of life.

PA has been shown to be influenced by several factors such as psychosocial aspects with motivation due to a peer group or family members, as well as prescriptions of healthcare professionals and others with a high impact on intrinsic motivation.

Sports, exercise training, and physical activity have an across-organ positive effect on the whole body. Even in patients with CHD, this effect is outreaching, with an improvement in motor development and muscle function, a reduction in body weight and metabolic diseases, as well as an improvement in the quality of life in children and their parents. 

The heterogeneity of CHD underlines the need for individual training programs and adapted intensities. 

Nevertheless, there is a low but appreciable risk of sports-related complications; however, sports in general do not increase the overall risk of mortality; this may be relevant. Most adverse events happen with participation in competitive sports and not just during participation in leisure sports.

In general, there seems to be only a small group of patients with heart diseases which should be excluded, or the risks should at least be discussed openly with parents and patients as in life-threatening arrhythmias (long QT syndrome, Brugada syndrome, catecholaminergic polymorphic ventricular tachycardia (CPVT), or other channelopathies), aortic dissection, or severe pulmonary hypertension.

A considerable approach might be a detailed past medical and family history, as well as exercise tolerance in a daily routine, and a clinical examination measuring blood pressure and ECG. For more complex or insecure patients, additional CPET or Holter-ECG might be useful. 

In order to increase the security of families and medical professionals, it might be useful to start training in a supervised home-based or institutional setting after adequate and individual pre-exercise assessment. Afterwards, individual interests and conditions should be taken into consideration when choosing a sport, including the intensity and weighing up the dangers.

Regarding the increase in quality of life when gaining normality and independence, the approach with training, based on target heart rates, gathered during CPET, seems to be desirable. This can increase the opportunity for patients with CHD to take part in sports clubs with their peers. 

## Figures and Tables

**Figure 1 children-10-00296-f001:**
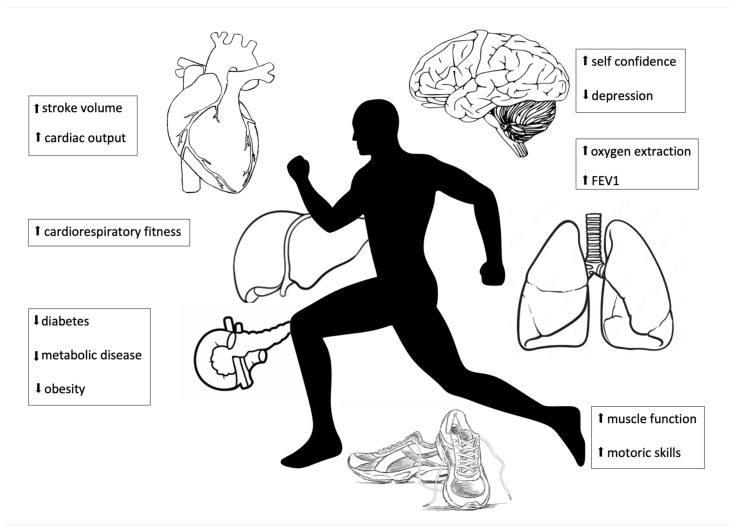
Effects of exercise training. 
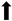
: improvement; 
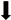
: reduction; FEV1: Forced expiratory volume.

**Figure 2 children-10-00296-f002:**
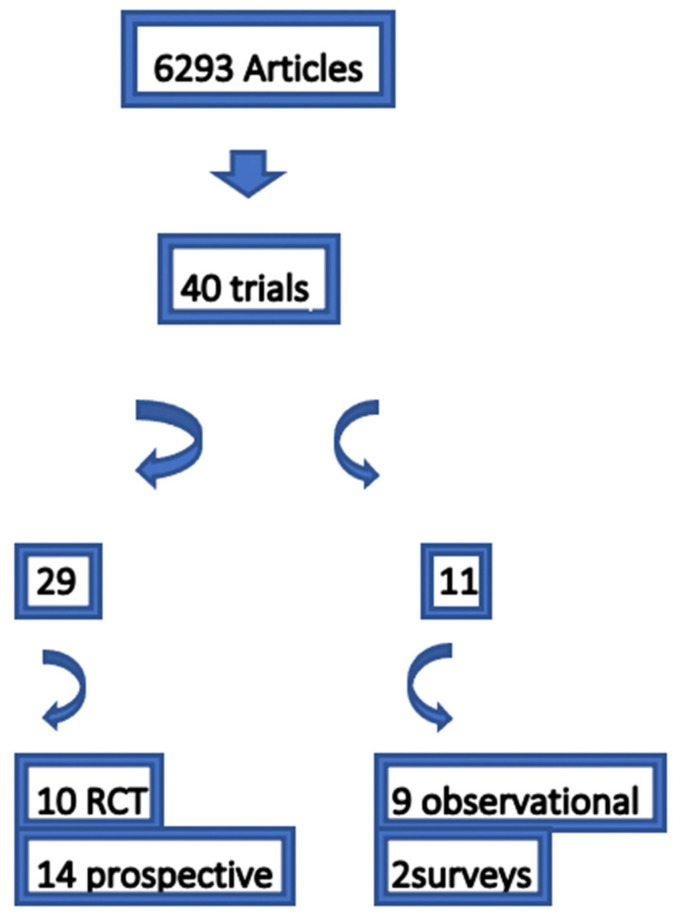
Included studies.

**Table 1 children-10-00296-t001:** Randomized controlled trials and prospective studies.

Study	Study Design	Diagnosis	Number of Participants	Age	Intervention	Control	Outcome	Results: Parameters with Significant Improvement	LOE
**Callaghan S et al. 2021** **[54]**	RCT	CHD, 100 males	163	5–10	One-day education session individual written exercise plan over the 4-month intervention period.	Usual care	CPET activity monitoring using an accelerometer.	Significant improvement in peak exercise capacity in the intervention group. Trend towards increased daily activity levels.	1
**Ferrer-Sargues FJ et al. 2021** **[55]**	Prospective interventional Study 11/2017–01/2020	CHD with reduced aerobic capacity	15	12.4–15.7	IMPROVE project (Initiative for Monitored Pediatric cardiac Rehabilitation Oriented by cardiopulmonary Exercise testing). IMPROVE was designed following the American College of Sports Medicine (ACSM). Cardiopulmonary 3-month rehabilitation program (CPRP), 2x/week for a total of 24 sessions. Sessions 70 min, endurance and strength-resistance training. Intensity was defined by the subject’s CPET, initially aiming for a heart rate (HR) near the first ventilatory threshold (VT1) HR and displacing this target frequency progressively towards the secondary ventilatory threshold (VT2) HR or a maximal HR of 75% of their peak HR in cases where the VT2 was not available. (a) Warm-up phase (5 min): diaphragmatic breathing, articular mobility exercises, and a light walk. (b) Endurance-training phase (20 min): treadmill, and a static bicycle, two min of warm-up, two min of cooldown. Ten to fifteen repetitions of each exercise, with a 20 s rest. Respiratory training (20 min): inspiratory muscle trainer threshold.	No	6MWT handgrip strength muscle strength arms and limbs.	Statistically significant improvement was observed in the subjects’ handgrip strength, biceps brachii, and quadriceps femoris strength, as well as triceps surae fatigue process with maintenance of the results six months after the intervention.	2
**Ferrer-Sargues FJ et al. 2021** **[56]**	Prospective interventional Study 11/2017–01/2020	CHD with reduced aerobic capacity	15	12–16	IMPROVE project (Initiative for Monitored Pediatric cardiac Rehabilitation Oriented by cardiopulmonary Exercise testing). IMPROVE designed following the American College of Sports Medicine (ACSM). Cardiopulmonary 3-month rehabilitation program (CPRP), 2x/week for a total of 24 sessions. Sessions 70 min, endurance and strength-resistance training. Intensity was defined by the subject’s CPET, initially aiming for a heart rate (HR) near the first ventilatory threshold (VT1) HR and displacing this target frequency progressively towards the secondary ventilatory threshold (VT2) HR or a maximal HR of 75% of their peak HR in cases where the VT2 was not available. (a) Warm-up phase (5 min): diaphragmatic breathing, articular mobility exercises, and a light walk. (b) Endurance-training phase (20 min): treadmill, and a static bicycle, two min of warm-up, two min of cooldown. Ten to fifteen repetitions of each exercise, with a 20 s rest. Respiratory training (20 min): inspiratory muscle trainer threshold.	No	Inspiratory muscle function. Spirometry/maximum static inspiratory and expiratory pressures. 6MWT handgrip strength, muscle strength, arms and limbs.	Increase in peripheral muscle function after a three-month, 24-session CPRP in children with CHD. Improvement persisted 6 months after the completion. Improvement of inspiratory muscles. Improved 6MWT. Results suggest CPRP improves respiratory muscle function and functional capacity, with lasting results.	2
**Kroll KH et al. 2021** **[57]**	Prospective interventional study	CHD (25 completed, 20 dropped out, 24 in progress) 25 (11 mental health diagnoses/ADHS)	25	7–24	“Steppin’ It Up” home-based year-long cardiac rehabilitation program. Four in-person visits occurring every 3 to 6 months. Garmin VivoFit2© Activity Monitor.		Exercise capacity was assessed by Progressive Aerobic Cardiovascular Endurance Run, (20-meter shuttle test run), patient functioning (social, emotional, school, psychosocial), patient general and cardiac-related quality of life, patient self-concept, and patient behavioral/emotional problems: HR-QoL/PQoL Piers’ Harris Children’s Self-Concept Scale Child Behavior Checklist	Progressive Aerobic Cardiovascular Endurance Run, significant increase from 5 to 10 median shuttles completed. PQoL in general not significantly improved, but cardiac-related quality of life improved Conversely, parents reported significant improvement in patients’ emotional functioning, social functioning, school functioning, psychosocial functioning, and total quality of life score. Overall self-concept, physical appearance and attributes, and freedom from anxiety improved	2
**Meyer et al.** **2021** **[58]**	RCT	CHD	70 35:35	10–18	3x/week 20 min web-based program over 24 weeks.	No training	Health-related physical fitness (HRPF) and health-related quality of life (HRQoL).	No improvement found.	1
**Blais AZ et al. 2020** **[59]**	Prospective interventional study	CHD	11	7–10	For 10 consecutive weeks, participants attended a once-weekly multi-sport program (Sportball ©). Each lesson focused on a different ball sport (i.e., basketball, volleyball, and soccer).	No	Questionnaire self-report; focus group sessions 40 min.	Themes: (a) motivation, (b) self-efficacy, (c) peer influences, and (d) family influences. Enjoyment of physical activity is a primary source of motivation, intrinsic motivation.	2
**Lopez et al. 2020** **[60]**	Prospective interventional study	CHD	104	9–16	ActiGraph accelerometer worn over the right hip during waking hours for 7 days.		Aortic pulse wave velocity (cm/s) was measured using standard two-dimensional echocardiography and Doppler ultrasound.	Higher levels of moderate-to-vigorous physical activity were associated with lower aortic pulse wave velocity.	2
**Sutherland N et al. 2018** **[61]**	Prospective, randomized trial 01/2010–06/2014	Fontan circulation (Home 11, Hospital 6)	17	12–19 15 ± 3	Home-based 2x1 h training/week (1 phone call + 1 visit week 4/5) Training: 5–10 min warm-up 20–30 min aerobic (target HR: 65–85% of HR at peak oxygen consumption); 20–30 min lower limb bodyweight resistance; 5–10 min cooldown.	Hospital-based: Training: 5–10 min warm-up 20–30 min aerobic (target-HR: 65–85% of HR at peak oxygen consumption) 20–30 min lower limb bodyweight resistance 5–10 min cooldown	CPET 6MWT QoL Reassessment after 8 weeks.	Oxygen consumption at anaerobic threshold increased from 19.3 ± 3.8 to 21.6 ± 6.0 mL/kg/minute. Peak oxygen pulse increased from 8.8 ± 2.5 to 9.5 ± 2.7 mL/beat. Total quality of life scale improved from 68 to 74%. Psychosocial health improved from 67 to 74%. 6MWT (521 ± 101 versus 569 ± 81 m).	1
**Altamirano-Diaz L et al. 2017** **[62]**	Prospective 05/2012–10/2015	CHD and with overweight or obesity 19 operated/non-operated	34	7–17	Bi-weekly nutrition and fitness counseling delivered via smartphone over 12 months; 30 min for each session: 15–20 min for counseling and 10–15 min for charting.		Anthropometry, body composition cardiorespiratory exercise capacity, and cardio-metabolic risk factors assessed at baseline, 6 months, and 12 months.	Statistically significant decreases in waist circumference (WC), body mass, waist-to-height ratio *z*-score were observed at 6 and 12 months in the operated group. Significant linear increase in lean body mass was observed in both groups. No significant difference between blood tests. VO_2_max at baseline, 6 months, or 12 months	2
**Hedlund et al. 2017** **[63]**	Prospective Interventional	Fontan	30	14.2 ±3.2	12-week endurance training subject’s organized physical exercise during an average school week. Duration in minutes was stated, and average perceived intensity was estimated using the Borg scale →individualized endurance training. The contract was to include 2 × 45 min of extra endurance training every week for 12 consecutive weeks, with maintained baseline activities such as physical education in school and other sports. The endurance training programs included sports such as running, jogging, skiing, cycling, riding, swimming, dancing, football, and so on. The purpose of the training program was to increase endurance training at a submaximal level with the aim to increase load gradually during the training program.	25 healthy	CPET 6MWT tests were repeated after a 12-week endurance training program and after 1 year. Questionnaire QoL Pediatric Quality of Life Inventory Version 4.0 questionnaires for children and parents.	CRF increased significantly in the control group but not in the patients.	2
**Jacobsen RM et al. 2016** **[64]**	Pilot study	Fontan (8 male)	14	8–12	Twelve-week moderate/high intensity home-based cardiac physical activity program. Home exercise routine. Three formalized in-person exercise sessions at 0, 6, and 12 weeks. A 45-minute home exercise routine of dynamic and static exercises. Fitbit Flex activity monitor	No	CPET HRQOL (child and parents) PedsQoL Shuttle Test Run to measure exercise capacity Fitbit Flex activity monitor.	Parents: overall HRQOL, physical function, school function, and psychosocial function. No measurable improvement in the patient-reported HRQOL. Objective exercise capacity significantly improved. VO_2_max from baseline to the 12-week session, improving from a mean of 41.8 to 42.3 mL/kg/min.	(2)
**Klausen et al. 2016** **[65]**	RCT	CHD	158	10–13	Paediatric Rehabilitation for Vanguard in Lifeskills (PReVaiL) All patients received 45 min of group health education and 15 min of individual counseling with their parents. Intervention: 52-week Internet, mobile application, and SMS-based program delivering individually tailored text messages to encourage physical activity. The program encompassed three main approaches: health education, tailored interactive text encouragements, and a personal exercise planning tool. The patients allocated to the eHealth intervention were sent health information and a new encouragement every week. Patients recorded exercise duration and type in a mobile application that translated intensity into virtual points, a system designed to provide motivation. Adherence to the eHealth program was assessed by patient registration of physical activities via the eHealth application for at least two consecutive weeks during the trial.	Without training	CPET	Of 81 patients in the intervention group, just 46 (57%) patients used the eHealth application for at least two consecutive weeks and completed both exercise tests. Only eight (10%) patients were active users during the last week of the intervention. Just 57 (70%) of the patients in the intervention group adhered to the intervention using the eHealth application for at least two consecutive weeks. PA and VO_2_max did not change significantly. Few patients used the app.	1
**Duppen et al. 2015** **[66]**	RCT	TOF Fontan	56 37 control 93	10–25	Twelve-week standardized aerobic exercise training program. Training program consisted of 3 1-hour exercise sessions per week The exercise sessions consisted of 40 min aerobic dynamic cardiovascular training, 10 min warming up, and 10 min cooling down. HR-monitor (aim to train submaximal level–baseline HR +60/70%)	Care as usual	CRF (CPET 2 weeks before and 2 weeks after training pro gram Stroke volume (cardiac MRI) PA (accelerometer)	VO_2_max increased in the exercise group by 5.0% (1.7 ± 4.2 mL/kg per minute) but not in the control group (0.9 ± 5.2 mL/kg per minute). Workload increased significantly in the exercise group compared with the control group (6.9 ± 11.8 vs. 0.8 ± 13.9 W). →in TOF, CRF increased (VO_2_max, O_2_pulse). in Fontan, it did not. PA level did not significantly raise after training.	1
**Dulfer K et al. 2014** **[67,68]**	RCT	TOF/Fontan	93	10–25	Twelve-week period 3x/week 1 h standardized exercise training. Training: 10 min warm-up, 40 min aerobic dynamic cardiovascular training, 10 min cooldown.	Care as usual	Web-based questionnaires and interviews by phone At baseline and after 12 weeks. Sports enjoyment and leisure-time spending. 2. Web-based age-appropriate HRQoL questionnaires 3. Psychological follow-up: phone interview, web-based questionnaire 0 + 3 months follow-up, focus on behavioral and emotional problems.	Exercise training decreased passive, but not active, leisure-time spending. It did not influence sports enjoyment. 2. children, aged 10–15 years in the exercise group improved significantly in self-reported cognitive functioning and parent-reported social functioning. Youngsters aged 16–25 years did not change their HRQoL. Fewer negative emotions than healthy peers. Less bodily pain, less role limitation due to physical limitations, and less role limitation due to emotional problems In contrast, they reported decreased general health. Patients in the control group had higher sustained scores on bodily pain. 3. No effects on behavioral and emotional problems.	1
**Longmuir PE et al. 2013** **[69]**	RCT	Fontan (36 male)	61	5.9–11.7	Twelve-month, parent-delivered home training programs to enhance physical activity, motor skills, fitness, and activity. Static and dynamic exercise, play-based physical activity. Monthly, specific written/illustrated instructions for four 1-week sessions 1.5–2.0 h/week	12-month, parent-delivered home education programs. Educational activities (e.g., games, stories, Web sites). Families received healthy lifestyle information, such as healthy eating, injury prevention, and activity benefits Accelerometry	MVPA baseline, 6, 12, 24 months gross motor skills, fitness, and activity attitudes	Gross motor skill was significantly greater at the end of the 2-year study period for both intervention groups combined. MVPA at 2 years was significantly greater.	1
**Morrison et al. 2013** **[70]**	RCT	CHD	143	12–20	The activity day was conducted as a motivational interview-style group exploring the nature of motivation towards exercise using small groups and visualization techniques. The concept of motivation was introduced. Participants completed a motivational rating sheet assessing their own feelings towards the importance of exercise, confidence, and readiness with regard to increasing their activity presession and postsession. Each participant was seen individually, and suggestions were discussed for ways to increase their activity over the next 6 months in a manner suitable for their diagnosis. They were also given a written exercise training plan to implement at home. Each participant was contacted once a month to check on progress with their exercise plan and discuss any problems.	No exercise	MVPA accelerator CPET endurance reassessment after the 6-month intervention period	Increased MVPA in intervention group. This study is unique in that it employed psychological methods to maximize the impact of its interventions. Consideration for issues, such as maintenance of activity, psychological well-being, and promotion of good lifestyle choices.	1
**Moalla et al. 2012** **[71]**	RCT	CHD	18	12–15	Individualized 12-week aerobic cycling training group (TG)	No intervention	Effect of training on peripheral muscular performance and oxygenation. Maximal voluntary contraction (MVC) and endurance at 50% MVC (time to exhaustion, Tlim) of the knee extensors were measured before and after training. During the 50% MVC exercise and recovery, near-infrared spectroscopy (NIRS) was used to assess the fall in muscle oxygenation, i.e., deoxygenation (DmO_2_ ) of the vastus lateralis, the mean rate of decrease in muscle oxygenation, the half time of recovery (T1/2R), and the recovery speed to maximal oxygenation (RS).	After training, significant improvements were observed in TG for MVC (101.6 ± 14.0 vs. 120.2 ± 19.4 N·m) and Tlim (66.2 ± 22.6 vs. 86.0 ± 23.0 s). Increased oxygenation (0.20 ± 0.13 vs. 0.15 ± 0.07 a.u.) and faster mean rate of decrease in muscle oxygenation were also shown after training in TG (1.22 ± 0.45 vs. 1.71 ± 0.78%·s^−1^). Moreover, a shorter recovery time was observed in TG after training for T1/2R (27.2 ± 6.1 vs. 20.8 ± 4.2 s) and RS (63.1 ± 18.4 vs. 50.3 ± 11.4 s). A significant relationship between the change in DmO_2_ and both MVC (r = 0.95) and Tlim (r = 0.90) in TG was observed	1
**Moalla et al. 2012** **[72]**	Prospective interventional Study	CHD	25	13.5 ± 1.8	Three-day multi-sports camp	Healthy controls	The perceived health status was measured using the Child Health Questionnaire–Child Form, CHQ-CF87, completed by the child at the start of the camp (T1), at the end of the camp (T2), and 3 months after the camp concluded (T3). Habitual physical activities were assessed by means of a modified version of the Baecke questionnaire, which was completed by one of the parents at T1 and T3.	Improvement during camp attendance. No change in habitual activity afterwards.	2
**Müller et al. 2012** **[73]**	Pilot study	CHD 04/2007–07/2011	14	4–6	Three-month low-dose motor training program of 60 min once per week.		Motor developmental test MOT 4–6 months before and after 3 months.	Delayed motor development significantly increased motoric skills	(2)
**Stieber NA et al. 2011** **[74]**	Prospective interventional study	toddler CHD (post-op ASO or SCPC) (5 female)	20	12–26 months	Ten weeks (5 two-week sessions), play-based, parent-delivered rehabilitation program on gross, fine, and visual motor functions. The total daily time requested was 20 min: 10 min for each of the two motor development goals identified for that 2-week period. Parents were contacted biweekly.	No	Peabody developmental motor scale-version 2 (PDMS-2) before and after the 10 week + video SCPC scored lower at baseline.	No significant differences between the baseline and follow-up age-adjusted scores in either group. Rehabilitation program enables post-SCPC children to increase their rate of development to an age-appropriate rate.	3
**Beek et al. 2010** **[75]**	Observational study	TGA switch	17	12.1 ± 2	Pedometer and diary.	20	Seven-day period using a pedometer and diary questionnaire was used to assess physical activity participation and overprotection	No significant differences in physical activity pattern or overprotection.	3
**Brassard P** **et al. 2006** **[76]**	Pilot study	Fontan	7	16 ± 5	Eight-week aerobic and resistance training program at the Pavillon de Prévention des Maladies Cardiaques de l’Hôpital Laval (n = 2) or at home (n = 3). Aerobic training was individually prescribed to allow the subjects to work progressively at 50–80% of their VO_2_ peak throughout the 8 weeks.	Control group of Fontan patients (n = 4) performed their normal activities without exercise training. Healthy (7)	CPET ergometer skeletal muscle function evaluation ergoreflex contribution	Fontan exercising group: no significant change in absolute or relative VO_2_ peak and in skeletal muscle strength. Neuromuscular function was positively influenced by exercise training. Lower ergoreflex contribution to absolute values of systolic blood pressure resulted from exercise training. An 87% reduction in the ergoreflex contribution to absolute values of diastolic blood pressure ( *p* = 0.2) and a 68% reduction in relative values of diastolic blood pressure ( *p* = 0.36) in the Fontan exercising group was observed.	2
**Moalla 2006** **[72]**	Prospective interventional Study	CHD	9	13.5 ± 1.8	Maximal volunteered contraction (MVC) and endurance at 50 % of MVC (time to exhaustion, Tlim) of the knee extensor were measured.	14 healthy	Near-infrared spectroscopy (NIRS) was used to evaluate StO_2_ and BV in vastus lateralis. The drop in muscle oxygen saturation (DmO_2_), half-time of recovery (T1⁄2R), and recovery speed to maximal oxygen saturation (Rs) were analyzed.	Patients with CHD showed lower MVC and Tlim than control children. StO_2_ and BV values in both groups were similar at rest and decreased at the onset of contraction. DmO_2_ was larger in patients, which reflected pronounced deoxygenation. During recovery, the patients exhibited a longer T1⁄2R and RS than control children. We concluded that reduced strength and endurance in patients with CHD were associated with an impairment of StO_2_ and BV and a slower reoxygenation during recovery. Mismatch between oxygen delivery and oxygen consumption during exercise. Patients with CHD showed reduced skeletal muscle strength and endurance of the knee extensors in comparison with the control children. Parameters of NIRS recovered significantly faster in control subjects than patients with CHD.	
**Moons et al.** **2006** **[77]**	RCT	CHD (+14 healthy)	17	12–16	9CHD Training: 12 weeks, 3 sessions/week, 1 h. (1) 10 min warm-up; (2) 45 min interval training alternating 10 min active periods at an HR within ± 5 bpm to the VT (individualized intensity) and passive 5 min periods of pedaling against an unloaded charge; (3) 5 min recovery. Cycle ergometer, pulse monitor.	14 healthy 8 CHD	CPET 6MWT	No significant difference for FEV1/FVC, TLC, and FVC between the two groups. After training period, T-CHD group significantly increased their WD. After 12 weeks of training, the ratios of power output at VT/peak, HR at VT(HRVT)/HRmax, VO_2_ at VT(VO_2_VT)/VO_2_max, and VE at VT (VEVT)/VE max increased significantly in T-CHD.	1
**Rhodes et al.** **2006** **[30]**	Prospective interventional study	CHD	15	8–17	1 h session 2x/week for 12 weeks .	18 CHD without training	Restudied 6.9 ± 1.6 months after completion of the cardiac rehabilitation program (1 year after the precardiac rehabilitation study).	The cardiac rehabilitation patients’ exercise function did not change significantly over the 6.9-month period. Predicted peak oxygen consumption and peak work rate remained significantly superior to baseline, precardiac rehabilitation values. These changes were also associated with improvements in self-esteem, behavior, and emotional state. In contrast, among the control subjects, small but statistically insignificant declines in peak oxygen consumption and peak work rate were observed on the final exercise test compared with values obtained at baseline, 1 year earlier.	2
**Opocher et al. 2005** **[78]**	Prospective interventional Study	Fontan	10	7–12	The training program lasted for 8 months. Ten lessons were held at a local gymnasium in Padova, Italy, twice a week for the first 3 weeks and once a month for the next 4 months of the training program. The remaining part of the program was held at each child’s home twice a week for 30 to 45 min under the supervision of parents. Each child was given a wristband heart rate monitor and instructed to keep their heart rate in the prescribed aerobic range. The exercise level during training was designed to range from 50% to 70% of maximal oxygen consumption. Parents were requested to write the frequency and duration of each exercise session in a log. Moreover, we contacted the families monthly by phone to inquire about the children’s compliance with the training program and to encourage continued participation.		CPET	Max. oxygen pulse significantly increased training results in an improvement in an aerobic capacity. Important increase in maximal oxygen consumption as well as a decrease in the heart rate curve and an increase in the oxygen pulse curve during submaximal exercise. Reduced heart rates and improved cardiovascular efficiency during usual daily activities.	2
**Rhodes et al. 2005** **[30]**	Pilot study	CHD (mostly Fontan)	19	8–17	Rehabilitation sessions were conducted for 1 hour twice a week for 12 weeks at the clinic. 5 to 10 minutes of stretching exercise. 45 minutes of aerobic and light weight/resistance exercises. Activities included aerobic dance, step aerobics, calisthenics (sit-ups, crunches, jumping jacks, push-ups, etc.), kickboxing, and jumping rope. When the weather permitted, outdoor games such as capture the flag and relay races were conducted. Resistance exercises were performed with 3 and 5 lb barbells, light elastic bands, and cords. Games, rubber balls, music, and simple, age-appropriate prizes (e.g., baseball cards) were incorporated into the activities to promote enthusiasm and motivation. Attempts were also made to vary activities and to accommodate the moment-to-moment desires of the patients to optimize participation and interest. The last 5 to 10 min of each session were devoted to cooling down and stretching. Patients were also encouraged to exercise at home on at least two additional occasions per week. Heart rate was checked (manually) at the start of each session and on two or three additional occasions during each session. The patients were encouraged to exercise at an intensity sufficient to raise their heart rates to levels equal to that associated with the ventilatory anaerobic threshold.		CPET	VO_2_max increased. Oxygen pulse increased .	2
**Fredriksen et al.** **2000** **[79]**	Prospective interventional	CHD	129	10–16	Rehabilitation facility- or home-based training for 5 months, 2x/week. The subjects were introduced to swimming, football, volleyball, and also more general activities which had the purpose of facilitating strength, balance, coordination, flexibility, and stamina. Outdoor activities included downhill skiing, cross-country skiing, and hiking. The activities had one common feature; the intensity of the exercise should induce 65—80% of peak heart rate for at least half of the time spent in physical activity, using the peak heart rate assessed at the exercise test prior to training.	38 healthy	CPET +PA before 1–2 w after the end of intervention. Survey (psychosocial) (Youth self-report; child behavior checklist) Polar heart rate counter Activity monitor	Controls gained weight. VO_2_max(l/min) increased in exercise groups significantly but did not reach values of control group. PA also significantly increased. Control group had a higher VO_2_max at the beginning compared to the intervention group, decreased in follow-up. Decrease in withdrawal and somatic complaints. No change in anxiety/depression.	2
**Longmuir PE et al. 1990** **[53]**	Prospective interventional study	CHD	40	?	Three-month home exercise program (directly postoperative, included endurance activities and jogging, a special program these groups used before).	Healthy children	Questionnaire of PA child and parents Exercise and scoring identical to an earlier study (1985). Reassessment after 6 months.	PA improved in those of the healthy pairs and persisted in the 5-year follow-up.	2

**Table 2 children-10-00296-t002:** Observational studies and surveys.

Author	Study Design	Diagnosis	Number of Participants	Age	Intervention	Control	Outcome	Results	LOE
**Dean PN et al. 2015** **[80]**	prospective observational study	CHD	177	13–30			PedsQoL Questionnaire regarding PA/sedentary	Adolescents: 18.6% reported physical activity of at least 60 min 7 days per week, and 39.5% reported physical activity 5 or more days per week, compared with 28.7% and 49.5%, respectively, of the nationally representative sample. Majority CHD recreational or competitive sports participation. CHD commonly participate in competitive sports. Participation in competitive sports and physical activity associated with improved QoL and exercise capacity and decreased BMI.	2
**Majnemer A et al. 2019** **[81]**	Prospective observational study	CHD 39 male	80	15 years 8 months (1 year 8 months; range 11 years 5 months–19 years 11 months)			Children’s Assessment of Participation and Enjoyment (CAPE) outcome measure of leisure participation.	Participants exhibited impaired motor (43.5%), behavioral (23.7%), and cognitive (29.9%) development. The most intense participation was in social (mean [SD] 3.3 [0.99]) and recreational (2.9 [0.80]) activity types on the CAPE. Male sex (*p* < 0.05) and younger age were associated with greater physical activity (<15 years: 1.87; ≥15 years: 1.31, *p* < 0.05). Greater engagement in social activities was related to better cognition (r = 0.28, *p* < 0.05), higher motor function (r = 0.30–0.36, *p* < 0.01), and fewer behavioral difficulties (r = 0.32 to 0.47, *p* < 0.01). Cognitive ability (r = 0.27, *p* < 0.05), dexterity and aiming/catching (r = 0.27–0.33, *p* < 0.05), and behavior problems (r = 0.38–0.49, *p* = 0.001) were correlated with physical activity participation. Persistence in tasks, an aspect of motivation, correlated with physical (r = 0.45, *p* < 0.001) and social activity involvement (r = 0.28, *p* < 0.05).	2
**Chen CW et al. 2015** **[82]**	Observational study	CHD	413	12–20			Physical self-description questionnaires and three-minute step tests.	The male participants had significantly greater scores in overall physical self-concept, competence in sports, physical appearance, body fat, physical activity, endurance, and strength.	
**Arvidsson D et al. 2009** **[83]**	Observational study		45 34	9–11 14–16			PA during seven consecutive days was assessed using the Acti- Regò activity monitor (PreMed AS, Ytre Enebakk, Norway), CPET.	PA levels similar to controls. most did not achieve the recommended 60 min of daily MVPA. Statistically significant associations between PA level, time spent on MVPA and aerobic fitness were found in girl patients, and between time spent on MVPA and aerobic fitness in younger control boys.	2
**O´Byrne ML et al. 2018** **[84]**	Retrospective cross-sectional trial	TOF TGA Fallot	253	8–18 13.1			Association between body mass index and duration of habitual exercise (measured by questionnaire).	Increased habitual activity associated with lower body mass index.	3
**Longmuir PE et al. 2021** **[85]**	Retrospective	CHD 26 male	45	6–12.4			Separate parent (*n* = 3) and child focus groups (*n* = 3), 4–6 participants per group, facilitated the expression of personal opinions. Questionnaire/interviews regarding PA. Focus groups were 60–90 min. Interviews were 30 (children) to 60 min (parents). Audio recordings.	Children with CCHD and their parents reported interactions. Social environment at home, social health environment, and social and physical environments related to peer interactions at school/daycare .	3
**Ray TA et al. 2011** **[86]**	observational study	CHD 51 males	84	10–14			Self-reported measures and clinical data. Previous day physical activity recall clinical data (BMI, etc.).	Low rates of physical activity and a higher obesity rate. Physical activity and body mass index were not significantly correlated.	3
**Kim HJ et al. 2018** **[87]**	Prospective observational study 11/2015–02/2016	CHD 61 Male	111	13–18			CPET global physical activity questionnaire (GPAQ) HRQOL/PedsQoL.	Boys more likely to participate in total physical activity (*p* = 0.06). Boys higher exercise capacity (*p* < 0.01). Self-reported and parent proxy-reported HRQOL were positively associated (*p* = 0.003; ß = 0.12; ß = 0.63, *p* < 0.001) with physical activity (ß = 0.16, *p* = 0.049) and exercise capacity.ß = 0.66, *p* < 0.001). Parent HRQOL differed. Improving exercise capacity to potentially enhance HRQOL in adolescents with complex CHD.	2
**O´Byrne ML et al.** **2017** **[88]**	single-center cross-sectional observational study with prospective and retrospective data 03/2012–12/2013	TGA Fontan Repaired biventricular	175	8–17.5			CPET Habitual exercise questionnaire.	VO_2_max lower in Fontan than normal cardiac anatomy (*p* < 0.0001) or TGA (*p* < 0.0001). Habitual exercise was not associated with VO_2_max in Fontan as compared to biventricular circulation.	3
**Knowles RL et al. 2014** **[89]**	Survey Cross-sectional study	CHD	477	10–14		464 control classmates		Participation in sports activities is positively associated with increased emotional well-being. Child self-report measures of QoL.	2
**Ray TD et al. 2010** **[90]**	Survey Cross sectional study	CHD 51 male	84	10–14			Self-efficacy instrument (SES; Dishman et al., 2002), the five physical activity items from the YRBS (Centers for Disease Control and Prevention [CDC], 1991.	Self-efficacy scores were moderately correlated with physical activity participation (r = 0.47; *p* < 0.001).	2

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
