# Peer review of "Effects of Sports, Exercise Training, and Physical Activity in Children with Congenital Heart Disease—A Review of the Published Evidence"

_children, 2023, doi:10.3390/children10020296_

Round 1
Reviewer 1 Report
Title: Effects of sports, exercise training and physical activity in children with congenital heart disease – a review of the published evidence
Article Type: Systematic Review
Summary
This article reviewed the effects of sport and exercise in children and adolescents with congenital heart disease. In this article, 10 randomized controlled trials, 14 prospective interventional trials, 9 observational trials, 2 surveys and 5 meta-analyses were reviewed. The results indicated that sports and exercise training appear to be effective and safe in congenital heart disease patients.
Evaluation
The topic of this study is interesting and the article is well written. For a systematic review article, the manuscript is quite straightforward and so I think that can be acceptable for publication. However, some points and suggestions should be addressed by the authors, in order to improve the quality of the manuscript.
Minor points and suggestions
Please add database’s name that you searched to the abstract, such as Medline, Scopus, …
Please add the date of your databases search, when you did it? Please add it to the abstract.
Why did you add meta-analysis? Usually, researchers add only original research work to the review papers and no other review and meta-analysis papers.
Please add a better-quality picture for figure 1.
Line 186. à 134 . what is it? And about the others. Line 196, etc.
In the table 1, number is about number of participants?
Please add the limitation of your study to the discussion section.
Reviewer 2 Report
The paper entitled “Effects of sports, exercise training and physical activity in children with congenital heart disease – a review of the published evidence” by Dold and colleagues is a review article with the aim to give a broad overview on the effects of higher physical activity-levels, participation in sports and training/rehabilitation programs with focus on children and adolescents with congenital heart disease (CHD). The main conclusion of the review is that physical activity, participation in sports and exercise training seems to have a wide range of positive impacts, starting with improving motoric skills and muscle function, bringing health benefits and finally do have a huge impact on quality of life. The paper is very interesting, well written and the authors reviewed in detail well-chosen literature from the last three decades. It will be very useful for clinicians taking care for children, adolescents and young adults with CHD, who underwent surgery correction, palliation or not.
I have only a few comments to make:
· In the paragraph “Data extraction” the authors mention Fig. 2 but it is not reported in the manuscript.
· In the paragraph “Results” the authors mention Table 1 and Table 2 but they are not reported in the manuscript.
· The acronym “CPVT”, which I can imagine is referred to catecholaminergic polymorphic ventricular tachycardia, is not explained in the text.
· The numbering of the citations should be more orderly. For example the first citation in the Introduction is better not be number 101.
In my opinion the paper can be accepted after minor revision.
